# Respiratory pathology in the *mdx/utrn*-/- mouse: A murine model for Duchenne Muscular Dystrophy (DMD)

**Marán Y. Hernández Rodríguez[1], Debolina D. Biswas[1], Aoife D. Slyne[2], Jane Lee[1], Evelyn Scarrow[1], Sarra M. Abdelbarr[1], Heather Daniels[3], Ken D. O'Halloran[2], Leonardo F. Ferreira[4], Charles A. Gersbach[3,5,6], Mai K. ElMallah[1,7,8]***

**1** Division of Pulmonary and Sleep Medicine, Department of Pediatrics, Duke University, Durham, North Carolina, United States of America, **2** Department of Physiology, University College Cork, Cork, Ireland, **3** Center for Advanced Genomic Technologies, Duke University, Durham, North Carolina, United States of America, **4** Department of Orthopedic Surgery, Duke University, Durham, North Carolina, United States of America, **5** Department of Biomedical Engineering, Duke University, Durham, North Carolina, United States of America, **6** Department of Surgery, Duke University Hospital, Durham, North Carolina, United States of America, **7** Department of Neurobiology, Duke University, Durham, North Carolina, United States of America, **8** Department of Cell Biology, Duke University, Durham, North Carolina, United States of America

\* mai.elmallah@duke.edu

## Abstract

Duchenne muscular dystrophy (DMD) is an X-linked devastating disease caused by a lack of dystrophin which results in progressive muscle weakness. As muscle weakness progresses, respiratory insufficiency and hypoventilation result in significant morbidity and mortality. The most studied DMD mouse model- the *mdx* mouse- has a milder respiratory phenotype compared to humans, likely due to compensatory overexpression of utrophin. *mdx/utrn*-/- mice lack both dystrophin and utrophin proteins. These mice have an early onset of muscular dystrophy, severe muscle weakness, and premature death, but the respiratory pathophysiology is unclear. The objective of this study is to characterize the respiratory pathophysiology and histopathology using whole body plethysmography to measure breathing and metabolism, diaphragm muscle functional analysis, histology, and immunohistochemistry. The *mdx/utrn*-/- mice have significant respiratory and metabolic deficits with respiratory insufficiency and hypoventilation when exposed to hypoxia and hypercarbia as early as 6 weeks of age. They also have significant diaphragmatic weakness and disrupted diaphragmatic structural pathology. The *mdx/utrn*-/- mice display respiratory dysfunction that mimics the DMD phenotype and therefore can provide a useful model to study the impact of novel therapies on respiratory function for DMD.

## Introduction

Duchenne muscular dystrophy (DMD) is a fatal X-linked neuromuscular disorder that affects approximately 1 in 3500 male births [1]. DMD is caused by mutations in the gene encoding dystrophin- a large protein that stabilizes muscle fibers [2–4]. Dystrophin resides along the sarcolemma that connects the skeletal muscle cytoskeleton to the transmembrane

**Data availability statement:** All relevant data are within the manuscript and its Supporting Information files

**Funding:** The project was supported by NIH NHLBI PROSPER 5T32HL160494-03 Grant (MHR), NIH NHLBI R01HL171282 (MKE), NIH R01AR069085 (CAG), USAMRMC MD140071 (CAG). ADS was funded by an Eli Lilly PhD scholarship with additional support from the Department of Physiology, University College Cork, Ireland. The funders had no role in study design, data collection and analysis, decision to publish, or preparation of the manuscript.

**Competing interests:** NO authors have competing interests

dystrophin-glycoprotein complex (DGC) and functions to prevent mechanical damage during muscle contraction and relaxation [5,6]. Deficiency of functional dystrophin causes stress in the myofibers during contraction and relaxation, resulting in chronic inflammation, degeneration, and necrosis. DMD is characterized by progressive muscular weakness, wasted striated muscle, fatty infiltration, and fibrosis [2,7]. Muscle weakness becomes apparent at approximately 2–3 years of age and this weakness is progressive with eventual respiratory deficits evident at around 10–12 years of age [8,9]. As the disease progresses, diaphragm and respiratory muscle weakness leads to respiratory function deterioration which clinically causes respiratory insufficiency and the need for ventilatory support [8,9]. If left untreated, this progressive respiratory muscle weakness leads to death from respiratory failure in the second to third decade of life [9–12]

Animal models of DMD have provided insight into the pathophysiology and progression of the disease. The *mdx* mice were the initial dystrophic model discovered and are the most studied DMD mouse model. This model has a nonsense mutation in exon 23 of the DMD gene [13,14]. However, the *mdx* mice often live up to 2 years of age similar to the lifespan of wild-type control mice [15]. This normal life span and less severe disease is most likely a result of compensation with utrophin in these mice. The utrophin protein is an autosomal analog of dystrophin localized in the skeletal muscular fibers [16]. In humans, utrophin is present at the sarcolemma during the fetal period and is replaced by dystrophin at birth [17]. Utrophin compensates for the absence of dystrophin in *mdx* mice, causing a milder phenotype of muscular dystrophy compared to humans [18–20]. The· *mdx/utrn*$^{-/-}$ mouse is deficient in both dystrophin and utrophin and exhibits severe disease. These mice show an early onset of muscular dystrophy, severe muscle weakness, and premature death [17]. Therefore, the *mdx/utrn*$^{-/-}$ mice are a more clinically relevant model and can play an important role in understanding the impact of novel therapies on respiratory function. In this study, we sought to examine the respiratory pathophysiology, diaphragm muscle functional analysis, and histopathology in the *mdx/utrn*$^{-/-}$ mice in order to provide clinically relevant respiratory outcome measures for the future study of novel therapies on respiratory function.

## Results

### Early onset and severe pathology with premature death of *mdx/utrn-/-* mice

Despite earlier studies that reported death in *mdx/utrn*$^{-/-}$ mice at approximately 20 weeks of age [17], the *mdx/utrn*$^{-/-}$ mice in our study died between 7–14 weeks (mean: 11 weeks) (Fig 1B). By 6 weeks of age, *mdx/utrn*$^{-/-}$ mice demonstrate a significant decline in weight compared to age-matched wild-type (WT) control mice (Fig 1C; *p = 0.003*). At 6 weeks of age, *mdx/utrn*$^{-/-}$ mice have significant kyphosis (Fig 1D; *p = 0.01*) and decreased forelimb strength (Fig 1E; *p = 0.007*) compared to WT mice.

### Respiratory insufficiency in *mdx/utrn-/-* mice

Respiratory function was assessed at baseline during normoxia ($FiO_2$: 0.21; nitrogen balance) followed by a respiratory challenge with hypoxia and hypercapnia ($FiO_2$: 0.10; $FiCO_2$: 0.07; nitrogen balance) by whole-body plethysmography (WBP) (Fig 2A). At baseline, *mdx/utrn*$^{-/-}$ mice have a significant increase in frequency *(p < 0.05* from 6–10 weeks) (Fig 2B) and reduced tidal volume (TV) (*p < 0.05* at all time points measured from 6–10 weeks, except at 8 weeks *p = 0.09)* (Fig 2C) when compared with WT mice. No significant differences are seen in minute ventilation ($V_E$), peak inspiratory flow (PIF), and peak expiratory flow (PEF) (Fig 2D-F) during normoxia. However, during the respiratory challenge, as early as 6 weeks of age, and as the disease progresses, *mdx/utrn*$^{-/-}$ mice have a decreased TV (*p < 0.05* from 6–10 weeks),

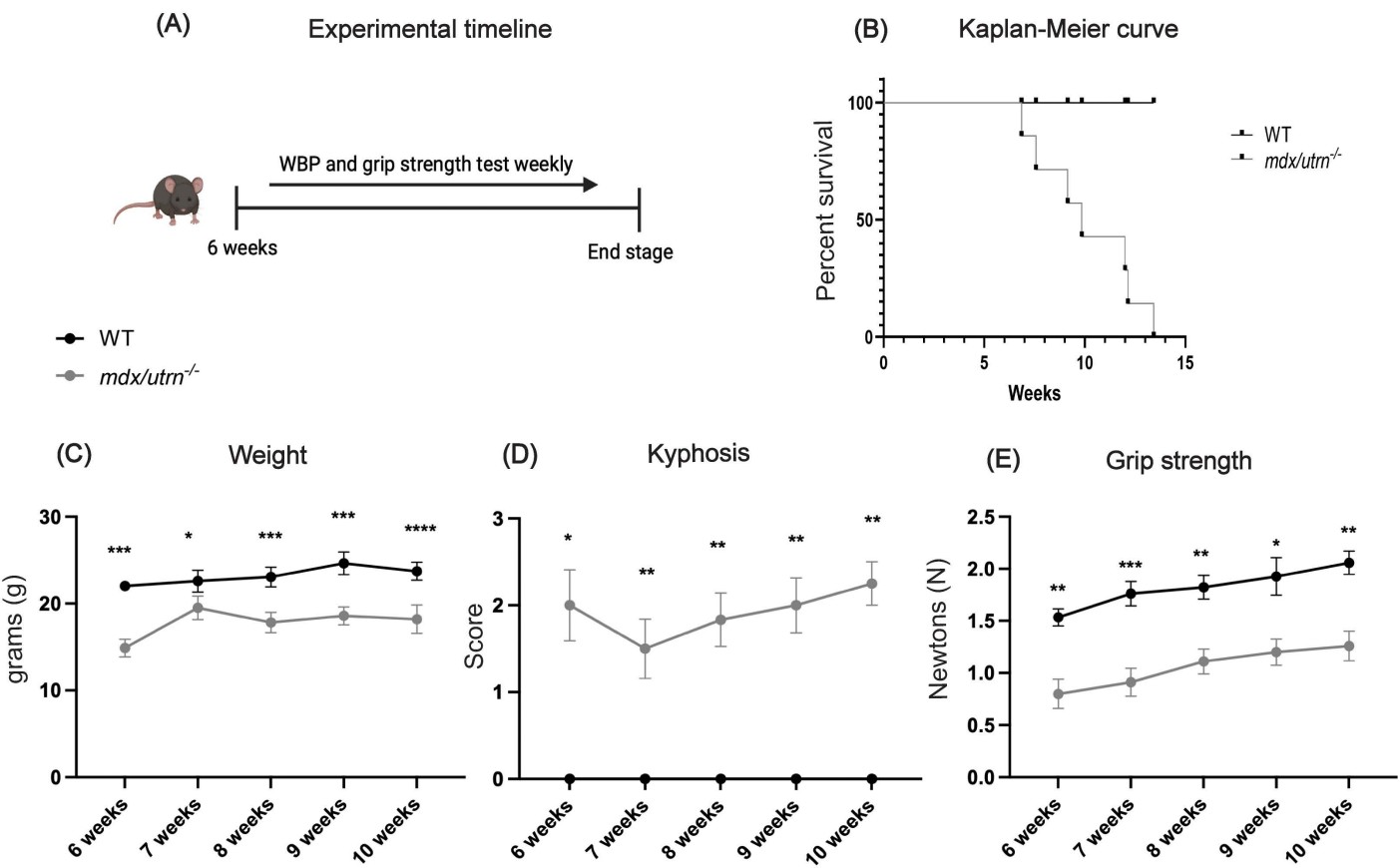

**Fig 1. Experimental design and characterization of *mdx/utrn-/-* mice.** (A) Experimental timeline of the study: Whole-body plethysmography (WBP) and grip strength testing were performed weekly starting at 6 weeks of age until disease end-stage. (B) Kaplan-Meier survival curve of male WT (n = 8) and *mdx/utrn-/-* (n = 8) mice. (C) The weight of the male WT (n = 8) and *mdx/utrn-/-* mice (n = 8) at indicated ages. (D) Scores for kyphosis in WT (n = 8) and *mdx/utrn-/-* mice (n = 8). (E) Grip strength of WT (n = 8) and *mdx/utrn-/-* mice (n = 8). Data are presented as mean±SEM. Statistical significance was determined using a mixed-model 2-way ANOVA; post hoc analysis using uncorrected Fisher's LSD * $P < 0.05$; **$P < 0.01$; ***$P < 0.001$, **** $P < 0.0001$.

$V_E$ ($p < 0.05$ from 6–10 weeks), PIF ($p < 0.05$ from 6–10 weeks) and PEF ($p < 0.05$ from 6–10 weeks) compared to WT mice (Fig 2C-F). No significant difference in frequency was seen between *mdx/utrn-/-* and WT mice during the respiratory challenge. These findings reveal that there is significant respiratory dysfunction in *mdx/utrn-/-* mice evident as early as 6 weeks of age and this progressively worsens with age.

## Hypoventilation in *mdx/utrn-/-* mice

To further assess if there is respiratory insufficiency and hypoventilation present, we measured carbon dioxide production ($V_{CO_2}$) (Fig 3A), and oxygen consumption ($V_{O_2}$) (Fig 3B). Minute ventilation ($V_E$) was normalized to $V_{CO_2}$ and $V_{O_2}$ to determine the ventilatory equivalent for carbon dioxide ($V_E/V_{CO_2}$) (Fig 3C) and oxygen ($V_E/V_{O_2}$) (Fig 3D). These measures were performed during normoxia and during a respiratory challenge with hypoxia and hypercapnia. During normoxia, there is no significant difference in $V_{CO_2}$, $V_{O_2}$, $V_E/V_{CO_2}$ and $V_E/V_{O_2}$ between *mdx/utrn-/-* and WT mice at 6–7 weeks or end-stage (~8–10 weeks of age) (Fig 3A–D). However, $V_{CO_2}$ and $V_{O_2}$ decreased in response to hypoxia and hypercapnia at 6–7 weeks ($V_{CO_2}$

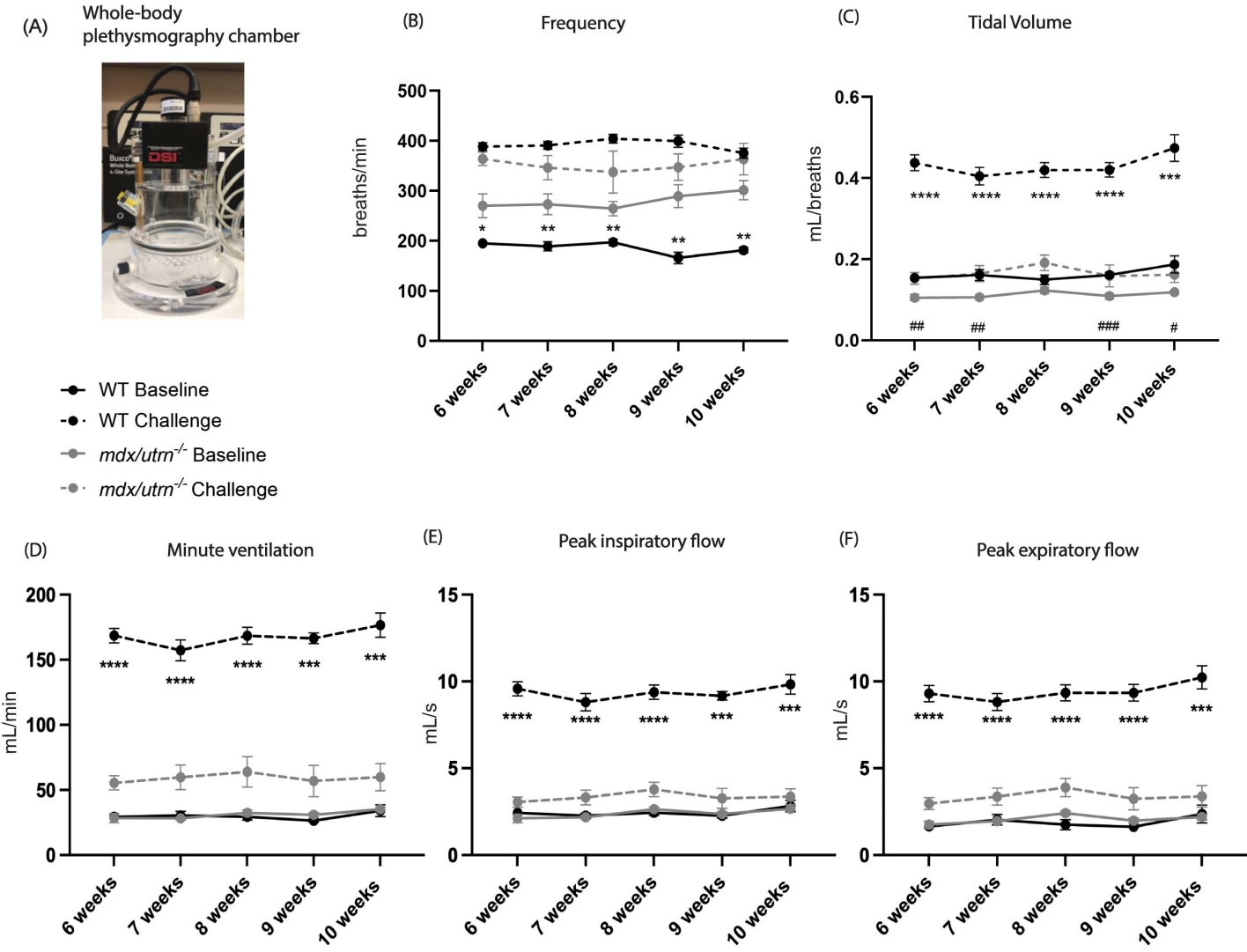

**Fig 2. Respiratory insufficiency in *mdx/utrn-/-* mice starting at 6 weeks of age.** (A) Whole-body plethysmography chambers are used to study respiratory parameters. (B-F) Measures of respiratory function in both WT (n = 8) and *mdx/utrn-/-* (n = 8) mice- frequency (B), tidal volume (C), minute ventilation (D), peak inspiratory flow (E) and peak expiratory flow (F) are analyzed during normoxia ("Baseline") and at a respiratory challenge with hypoxia and hypercapnia at the indicated ages. Data are presented as mean±SEM. Statistical significance was determined using a 2-way mixed-model ANOVA; post hoc uncorrected Fisher's LSD test. *#P < 0.05; ##P < 0.01; ###P < 0.001, #### P < 0.0001* during normoxia and * *P < 0.05; ** *P < 0.01; *** *P < 0.001, **** *P < 0.0001* during respiratory challenge.

$p < 0.0001$; $V_{O_2}$ $p < 0.0001$) and end-stage ($V_{CO_2}$ $p < 0.0001$; $V_{O_2}$ $p = 0.003$) (Fig 3C–D). $V_E/V_{CO_2}$ ($p = 0.0001$) and $V_E/V_{O_2}$ ($p = 0.0082$) in *mdx/utrn-/-* mice decreased in response to hypoxia and hypercapnia at end-stage, however, at 6–7 weeks, there is no significant difference in these parameters ($V_{CO_2}$: $p = 0.20$; $V_{O_2}$: $p = 0.79$) when compared with WT (Fig 3A–B). These findings imply that *mdx/utrn-/-* mice have impaired ventilation that is exacerbated at end-stage.

## Diaphragm muscle weakness in *mdx/utrn-/-* mice

To evaluate the involvement of diaphragm muscle dysfunction in impaired ventilation in *mdx/utrn-/-* mice, we examined the *ex vivo* functional capacity of the diaphragm muscle of wild-type

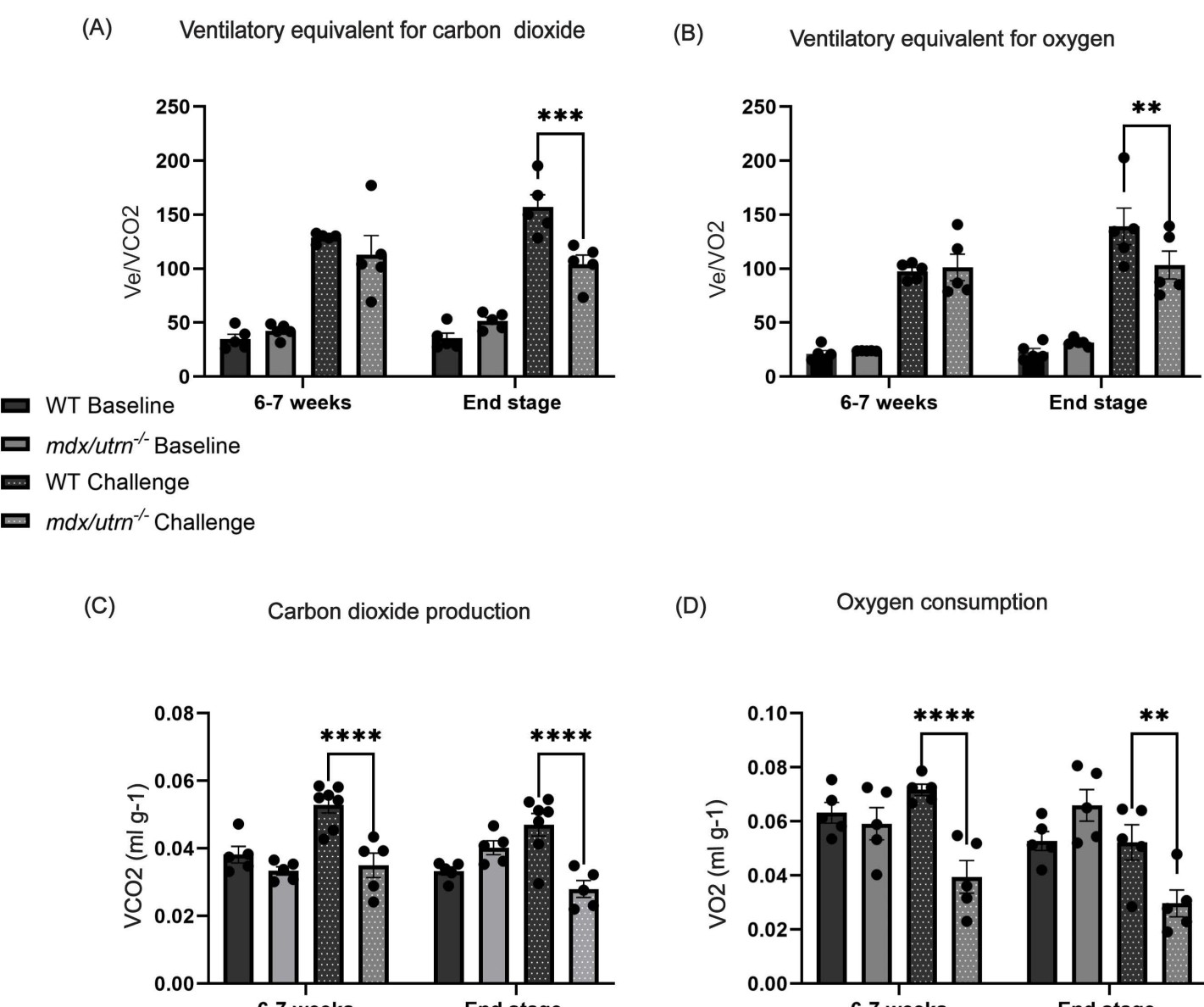

**Fig 3. Hypoventilation in the *mdx/utrn-/-* mice.** (A-D) Measures of respiratory metabolism in both WT (n = 5) and *mdx/utrn-/-* (n = 5) mice at 6–7 weeks of age and disease end stage- ventilatory equivalent for carbon dioxide ($V_E/V_{CO2}$) (A), ventilatory equivalent for oxygen ($V_E/V_{O2}$) (B), carbon dioxide production ($V_{co2}$) (C), and oxygen consumption ($V_{o2}$) (D) are analyzed during normoxia ("Baseline") and at a maximal respiratory challenge with hypoxia and hypercapnia. Data are presented as mean±SEM. Statistical significance was determined using a 2-way mixed-model ANOVA; post-hoc uncorrected Fisher's LSD test * $P < 0.05$; ** $P < 0.01$; *** $P < 0.001$, **** $P < 0.0001$.

and *mdx/utrn-/-* mice at 6 weeks of age, across a range of stimulation frequencies. Representative original traces for WT and *mdx/utrn-/-* diaphragm force-frequency relationship are shown in Fig 4A. The *mdx/utrn-/-* mice diaphragm exhibited a significantly depressed specific force generation compared to wild-type across a stimulus range of 50–300Hz (*p < 0.0001*) (Fig 4B). In addition, muscle power was decreased in *mdx/utrn-/-* mice (*p < 0.05*) (Fig 4C) when compared to WT. These findings indicate profound weakness in the primary inspiratory muscle of the *mdx/utrn-/-* mice as early as 6 weeks of age.

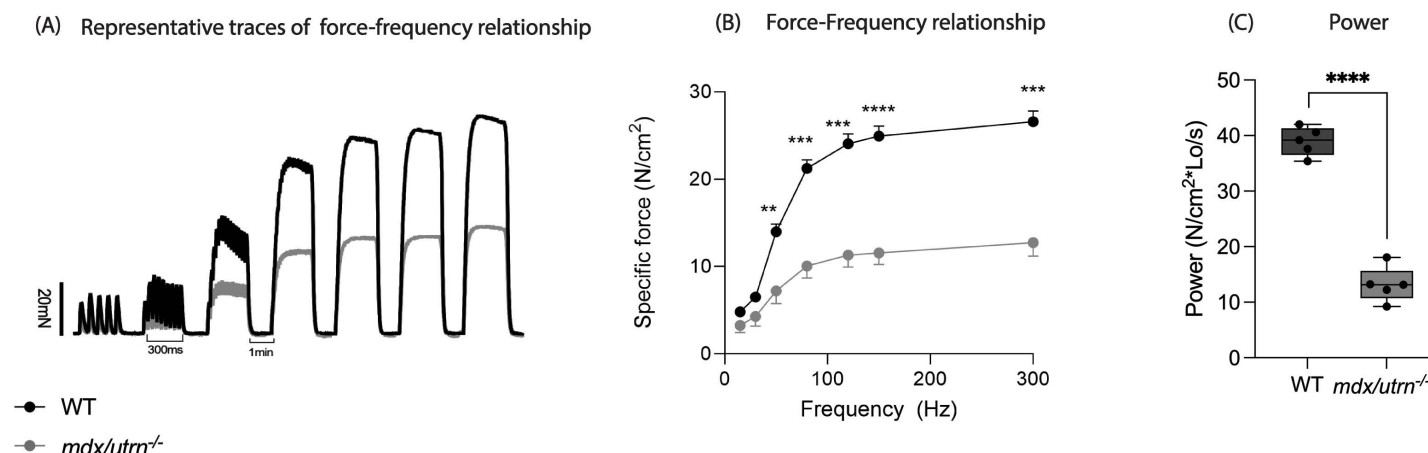

**Fig 4. Diaphragm muscle weakness in the *mdx/utrn*-/- mice.** (A) The representative traces of *ex vivo* diaphragm muscle force-frequency relationship for wild-type and *mdx/utrn*-/- mice. (B-C) Diaphragm function at 6 weeks of age in both WT (n = 5) and *mdx/utrn*-/- (n = 5) mice- diaphragm- force-frequency relationship across a broad stimulus range (50–300 Hz) (B) and power of diaphragm at 35% of maximal tetanic force (C). Data are presented as mean±SEM. Statistical significance was determined using a two-way RM ANOVA; post-hoc uncorrected Fisher's LSD test (B) and unpaired Student's *t*-test (C) * $P < 0.05$; **$P < 0.01$; ***$P < 0.001$, **** $P < 0.0001$.

## Histopathology of *mdx/utrn-/-* mice

To evaluate the histopathology of the muscles of *mdx/utrn*-/- mice, we performed hematoxylin and eosin (H&E) staining on the diaphragm, the main inspiratory muscle, the tongue, a muscle important for upper airway patency, and the tibialis anterior (TA) muscle, a peripheral muscle important for ambulation. The *mdx/utrn*-/- muscles displayed significant pathology as evidenced by increased cellular infiltrate and centralized nuclei in the diaphragm, tongue, and TA when compared to WT mice (Fig 5A–C). Infiltration of putative inflammatory cells and fibrosis (black arrows) and centralized nuclei (white arrows) indicative of muscle degeneration and regeneration of myofibers are observed in *mdx/utrn*-/- mice (Fig 5A–C). Centrally located nuclei were evident in approximately 35–40% of muscle fibers. These markers of muscle pathologies are not observed in WT mice. Next, using Sirius red staining, we investigated the accumulation of collagen in the diaphragm, tongue, and TA to assess fibrosis. We observed an increase in collagen deposition in *mdx/utrn*-/- mice compared to WT mice (Fig 6A–C) which suggests that there is significant fibrosis and muscle degeneration in *mdx/utrn*-/- mice compared to WT mice. Lastly, as expected, in WT mice, there is robust dystrophin expression in all muscle fibers in the diaphragm, tongue, and TA, whereas *mdx/utrn*-/- muscles lack dystrophin (Fig 7A–C).

## Neuromuscular junction analysis

Neuromuscular junctions (NMJs) in the diaphragm of WT and *mdx/utrn*-/- mice were analyzed. When stained for the post-synaptic membrane of NMJ using α-bungarotoxin, we found the *mdx/utrn*-/- mice have fragmented postsynaptic membranes (Fig 8A-B) compared to WT mice. Next, we analyzed the WT and *mdx/utrn*-/- diaphragms for the expression of post-synaptic genes *Musk* and *Chrna7* by qPCR. There is a significant increase in the expression of both *Musk (p = 0.008)* and *Chrna7 (p = 0.01)* in the diaphragm of *mdx/utrn*-/- compared to WT mice (Fig 8C-D).

## Discussion

Our primary findings are that *mdx/utrn*-/- mice have significant respiratory insufficiency, severe diaphragm muscle weakness and significant histopathology in the muscle and

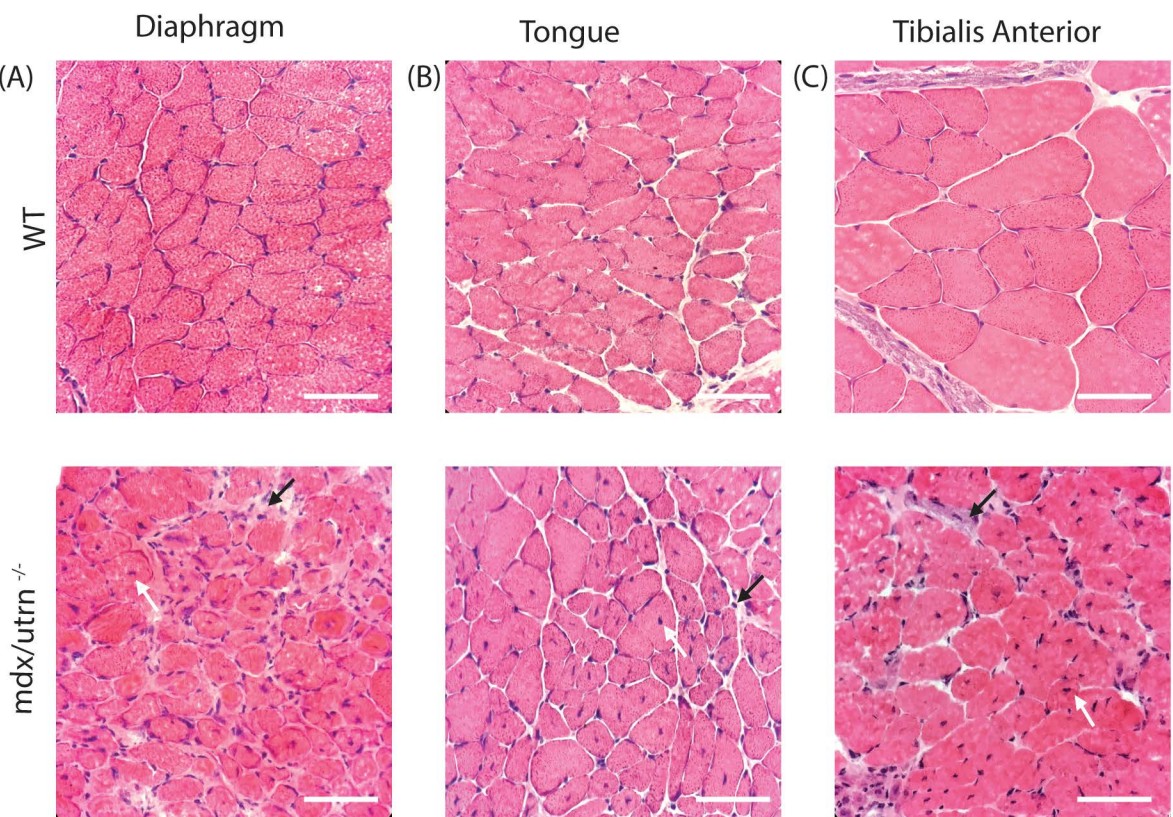

**Fig 5. Histopathology in the *mdx/utrn-/-* mice.** (A-C) Representative images of hematoxylin and eosin stained diaphragm (A) and tongue (B) and tibialis anterior (C) of the end-stage *mdx/utrn-/-* and same-aged WT control mice. The black arrow indicates the infiltration of inflammatory cells, and centralized nuclei are indicated by the white arrow. Scale bars: 90 µm.

neuromuscular junction. The respiratory insufficiency is evident at baseline during normoxia and is exacerbated during a respiratory challenge with hypoxia and hypercapnia. The functional capacity of the diaphragm, a major inspiratory muscle, is profoundly impaired in *mdx/utrn-/-* mice at 6 weeks of age. The diaphragm, tongue, and tibialis anterior have myofiber degeneration with increased fibrosis, and the diaphragm exhibits fragmented post-synaptic membranes of the NMJ with a compensatory increase in RNA expression of post-synaptic markers.

## Respiratory pathology in DMD

In patients with DMD, respiratory function starts to decline at approximately 12 years of age and is secondary to inspiratory and expiratory respiratory muscle weakness [11,12,21]. The diaphragm is the major inspiratory muscle progressively impacted by the disease [21]. In DMD, diaphragm exertion decreases and fat fraction increases with age [22]. The respiratory muscle dysfunction continues to progress resulting in restrictive lung disease and reduced maximal inspiratory and expiratory pressures [7]. Restrictive lung disease causes a decrease in vital capacity and progressive respiratory insufficiency culminating in respiratory failure and death around the second decade of age [9]. 5-year survival is only 8% once the vital capacity falls below 1 liter [8]. In addition, in patients with DMD, progressive tongue weakness leads to

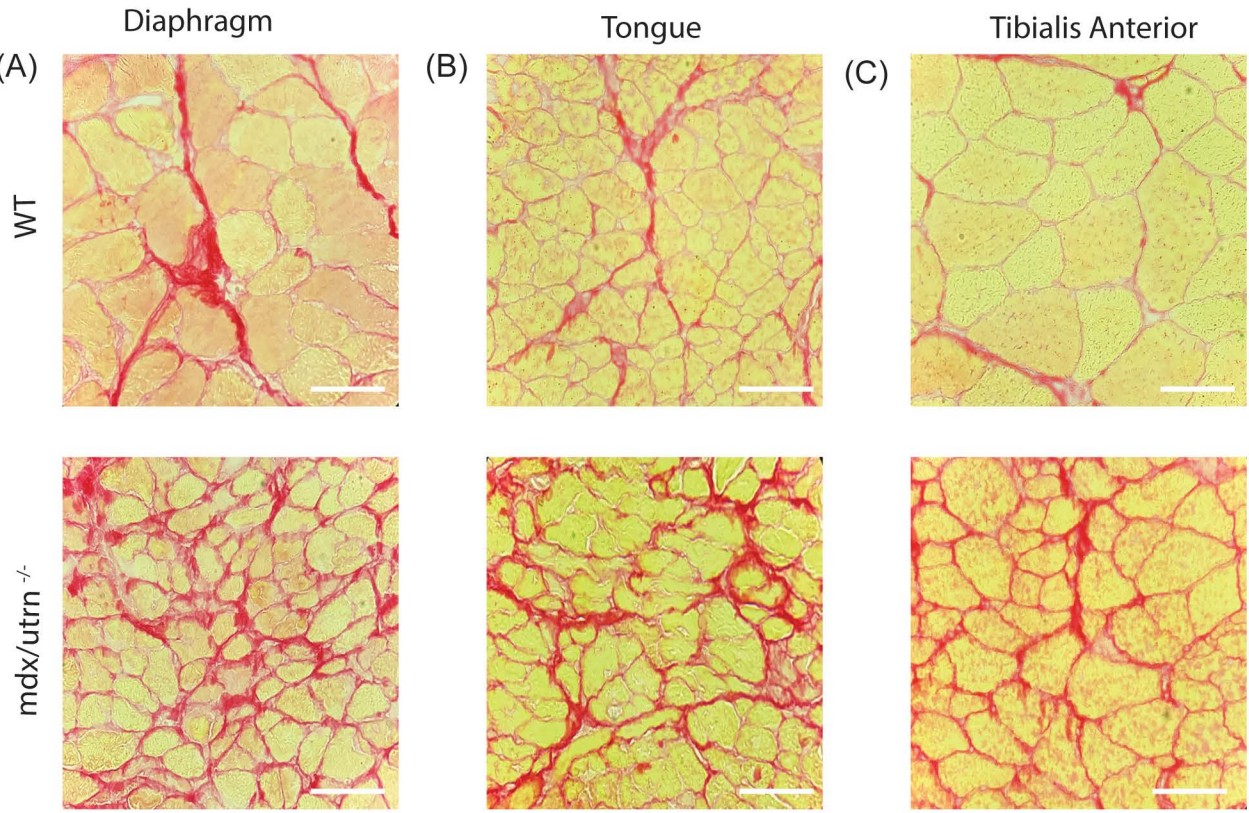

**Fig 6. Muscle fibrosis in the *mdx/utrn*-/- mice.** (A-C) Picro-Sirius red staining to measure fibrosis in the diaphragm (A), tongue (B) and tibialis anterior (C) of end-stage *mdx/utrn*-/- and age matched WT control mice. Scale bars: 90 μm.

obstructive sleep apnea, failure to control upper airway secretions, aspiration pneumonia and resultant hypoventilation, hypoxia and hypercapnia [23].

In the *mdx* mouse model of DMD, WBP has been extensively used to characterize respiratory pathology. However, to our knowledge, there is no respiratory characterization of the *mdx/utrn*[-/-] mouse model. We found that these mice cannot mount a robust respiratory response to a respiratory challenge with hypoxia and hypercarbia compared to WT mice as early as 6 weeks of age. Interestingly, our findings show that $CO_2$ production at normoxia was unaffected in *mdx/utrn*[-/-] mice. During normoxia, they also have a significant reduction in tidal volume that is compensated by increased breathing frequency, resulting in unaffected minute ventilation. However, when exposed to hypoxia and hypercapnia, *mdx/utrn*[-/-] mice show a decrease in $CO_2$ production, and $O_2$ consumption. In addition, *mdx/utrn*[-/-] mice have a reduced $V_E/V_{O2}$ and $V_E/V_{CO2}$ $O_2$ when exposed to hypoxia and hypercapnia. These data indicate that *mdx/utrn*[-/-] mice hypoventilate as early as 6 weeks of age and this worsens with disease progression. When comparing *mdx/utrn*[-/-] mice with other DMD mouse models, there are differences in the age of onset of respiratory dysfunction in the *mdx* mice. Some studies showed that *mdx* mice have respiratory dysfunction starting at 2–3 months of age [24–27], while Gayraud *et. al.* reported altered ventilation at 16 months in the *mdx* mice [28]. In

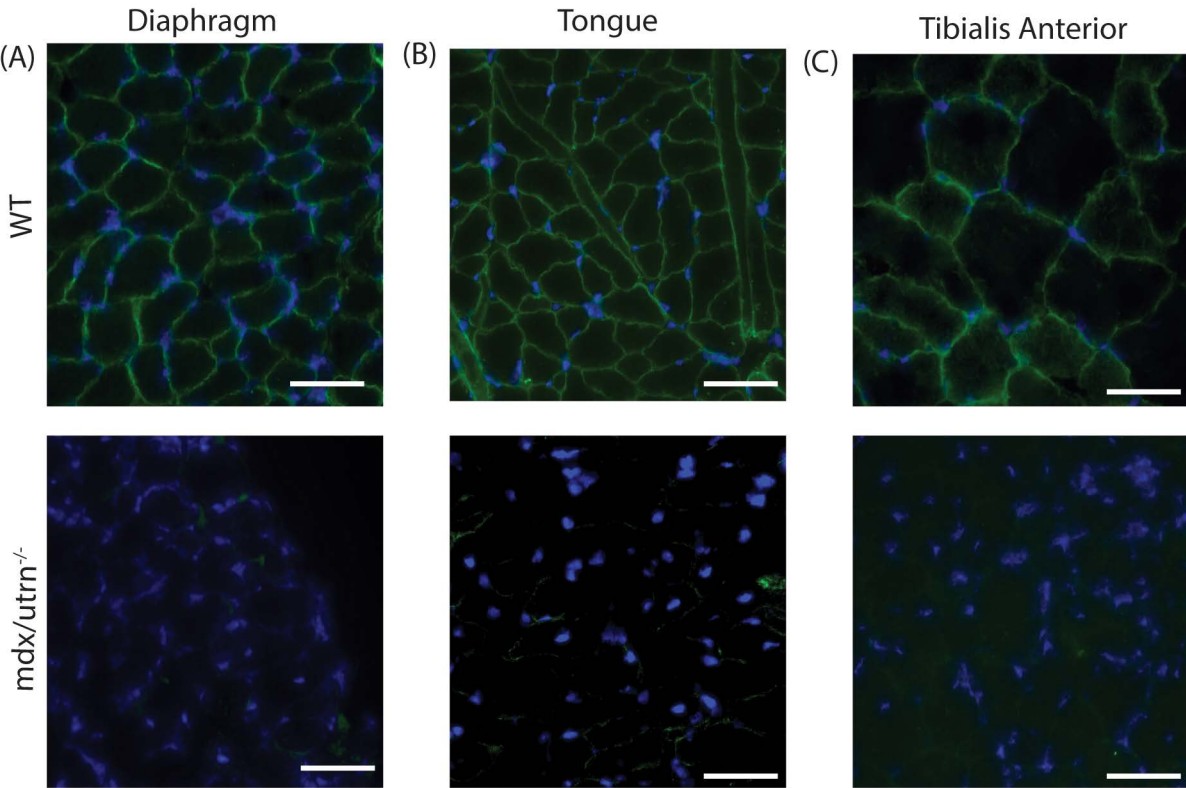

**Fig 7. Absence of dystrophin in the *mdx/utrn-/-* mice.** (A-C) Dystrophin staining of the diaphragm (A), tongue (B) and tibialis anterior (C) of end-stage *mdx/utrn-/-* and age matched WT control mice. Dystrophin (green) DAPI (blue). Scale bar: 90um.

addition, prior studies showed that *mdx/utrn*[+/-] mice have respiratory dysfunction much later than *mdx/utrn*[-/-] mice with findings of respiratory pathology at 3 months of age that worsens at 6 months of age [25]. Furthermore, our study confirms that there is significant diaphragm weakness in *mdx/utrn*[-/-] mice as early as 6 weeks of age. To our knowledge, this is the first study to assess diaphragm muscle function and confirm diaphragm weakness in *mdx/utrn*[-/-] mice. The diaphragm force-generating capacity in *mdx/utrn*[-/-] mice was reduced across a range of stimulation frequencies. The force increases with different stimulation frequencies in *mdx/utrn*[-/-] mice suggests that they are capable of increasing breathing, however ventilatory capacity is curtailed, which likely relates to impaired neural control of breathing. Severe diaphragm weakness in *mdx/utrn*[-/-] mice at 6 weeks of age may contribute to the decreased tidal volume at baseline and impaired ventilation when exposed to hypoxia and hypercapnia. In comparison, *mdx* mice have diaphragm weakness at 8 weeks of age, and ventilatory insufficiency is evident at 16 weeks of age [27,29]. Since the *mdx/utrn*[-/-] mice lack the expression of both dystrophin and utrophin, this results in early onset diaphragm weakness and disease progression with respiratory insufficiency and hypoventilation that closely mimics human disease.

## Muscle pathology in DMD

In the absence of dystrophin, muscle fibers undergo repetitive damage, which ultimately leads to myofiber replacement with fibrotic and adipose tissue [30]. Inflammation is needed for muscle repair but hyperactivation of the immune response in DMD is counterproductive and detrimental to the muscle, leading to fibrosis and impaired muscle repair [31]. Aberrant and

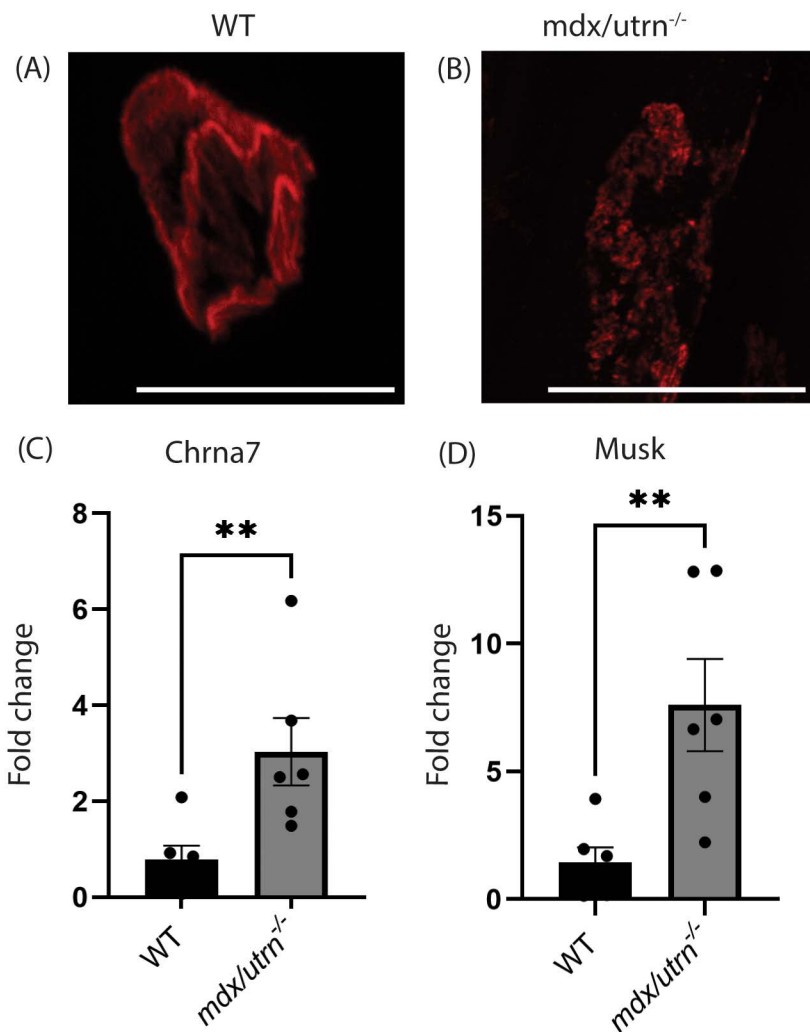

**Fig 8. Fragmented neuromuscular junction morphology in *mdx/utrn-/-* mice.** (A-B) The motor end plate of the diaphragm from WT (A) and *mdx/utrn-/-* (B) mice stained with α-bungarotoxin. (C-D) Expression of postsynaptic markers- *Chrna7* (C) and *Musk* (D) on NMJ in the diaphragm of the end-stage *mdx/utrn-/-* mice and age-matched WT control mice. Scale bars: 50 μm. Statistical significance was determined using unpaired Student's t-test * $P < 0.05$; ** $P < 0.01$; *** $P < 0.001$, **** $P < 0.0001$.

excessive pro-inflammatory signaling is observed in skeletal muscles of DMD patients from a young age and is hypothesized to be a major contributor to muscle weakness and disease progression [32,33]. Muscle remodeling and dysfunction are exacerbated by chronic inflammation. In the current study, we determined increased putative inflammatory cell infiltrate in the diaphragm and tibialis anterior muscles of *mdx/utrn*[-/-] mice. Although not directly compared in this study, previous studies have determined that *mdx/utrn*[+/-] mice display more exaggerated and dysregulated muscle immune responses than the conventional *mdx* mouse model [34]. Furthermore, we found a higher density of centrally nucleated fibers at the end stage of disease in the diaphragm, tongue, and tibialis anterior muscles of *mdx/utrn*[-/-] mice, which indicates muscle fiber repair and regeneration. Previous studies have determined that limb muscles of 2-month-old *mdx/utrn*[-/-] mice express a higher density of centrally nucleated fibers

than 12–20-month-old *mdx* and *mdx/utrn*[+/-] mice [35]. In addition, fibrosis occurs in muscle due to dysregulated repair processes that result from chronic muscle injury and inflammation, impairing muscle function. Fibrosis is a feature of various models of DMD [15,35–37]. We confirm that *mdx/utrn*[-/-] mice display increased collagen deposition in the diaphragm, tongue, and tibialis anterior muscles as early as 6 weeks of age. Previous studies directly comparing *mdx*, *mdx/utrn*[+/-] and *mdx/utrn*[-/-] mice have determined that at 8 weeks of age, the diaphragm muscles of *mdx/utrn*[-/-] mice express a higher percentage of collagen deposition than *mdx* and *mdx/utrn*[+/-] mice [38]. However, in the limb muscles, *mdx/utrn*[-/-] and *mdx*/utrn[+/-] mice display both an earlier onset and more severe pathology than *mdx* mice [38]. Overall, *mdx/ utrn*[-/-] mice exhibit severe pathology that more closely mimics human disease.

### NMJ pathology in DMD

NMJ impairments have also been reported in mouse models of DMD, including the *mdx/ utrn*[+/-], *mdx/utrn*[-/-] and *mdx* mice [39–44]. The DGC accumulates in the postsynaptic membrane of the NMJs and disruption of one of the proteins might affect the stability and function of the NMJ [39]. Both dystrophin and utrophin are present in the NMJ and are required for end plate maintenance [45,46]. In our study, we confirm fragmentation in the post-synaptic membrane. Previous studies have shown fragmentation of acetylcholine receptor clusters in the diaphragm in *mdx*, *mdx/utrn*[+/-], and *mdx/utrn*[-/-] mice, with *mdx/utrn*[-/-] mice displaying more severe disruption [44]. Furthermore, we were able to assess the alteration of postsynaptic RNA expression of *Musk* and *Chrna7*. *Musk* plays a role in the stability, maturation, and remodeling of the NMJs [47] and regulates the agrin-binding signalling which is involved in microtubule capture and stability of the NMJs [48]. In our study, both postsynaptic proteins, *Musk* and *Chrna7,* were upregulated as previously seen in *mdx* mice [49–51]. We speculate that the increase in RNA expression of *Musk* and *Chrna7* is a compensatory mechanism to preserve the stability and function of the NMJ [51]. The NMJ impairment most likely exacerbates the respiratory insufficiency observed in *mdx/utrn*[-/-] mice.

## Conclusion

This study highlights the severe respiratory pathology that exists in the *mdx/utrn*[-/-] mice observed as early as 6 weeks of age and continues to worsen with disease progression. Respiratory pathology is due to restrictive lung disease caused by muscle weakness and fibrosis. In addition, tongue weakness and fibrosis cause upper airway instability contributing to respiratory insufficiency and hypoventilation. This study is limited by the fact that the *mdx/utrn*[-/-] mice were not studied until they were 6 weeks of age. We hypothesize that respiratory pathology is present earlier than this time point. Future research is needed to examine the respiratory pathology of *mdx/utrn*[-/-] mice at earlier time points to assess the optimal timing of novel gene therapies to treat the disease. Overall, *mdx/utrn*[-/-] mice provide a useful model to study the impact of novel therapies on clinically relevant respiratory function outcome measures for DMD.

## Materials and methods

### Animals

The *mdx/utrn*[-/-] male mice were used in this study and compared with control male mice: the C57Bl/6J WT mice. Male *mdx/utrn*[-/-] and WT mice were generated in Dr. Charles Gersbach's laboratory. Utrn[tm1kEd]Dmd[mdx]/J mouse carries a neomycin cassette in exon 7 of the mouse utrophin gene, preventing Utrn expression [17]. The *mdx/utrn*[-/-] mice used in this study were generated by backcrossing Utrn[tm1kEd]Dmd[mdx]/J mouse from the Jackson Laboratory (Stock No. 014563, Bar Harbor, ME, US) [17] to C57BL/6 J mice. The mice were obtained at 5–6

weeks of age from the Dr. Charles Gersbach Laboratory (Duke University, Durham, North Carolina, USA). The mice were housed at the Duke University Division of Laboratory Animal Resources on a 12-h light/dark cycle with *ad libitum* access to food and water. Beginning at 8 weeks of age, mice were provided wet Purine Rodent Laboratory Chow (Purina, St Louis, MO, USA) and HydroGel (Clear $H_2O$, Portland, ME, USA) to supplement their daily food and water supply. Mice were monitored daily to assess for termination criteria: inability to rise by themselves, severe labored breathing, and decrease in body weight by 15% or more within a week. All animal experiments were approved by the Institutional Animal Care and Use Committee (IACUC) at Duke University.

## Respiratory and metabolism analysis

Respiratory function was assessed weekly using WBP starting at 6 weeks of age until 10 weeks of age or until disease end-stage (n = 8 per genotype), as previously described [51–58]. Unanesthetized and unrestrained mice were placed in a calibrated Plexiglas chamber (DSI, St. Paul, MN). Respiratory measurements were recorded and analyzed using FinePointe Software. Respiratory data was collected for 1.5 hours during normoxia ($FiO_2$: 0.21; $N_2$ balance), and 5 minutes of breathing at rest was selected as the baseline. The mice were then exposed to a 10-minute hypercapnic and hypoxic respiratory challenge ($FiCO_2$: 0.07, $FiO_2$: 0.10; $N_2$ balance), then returned to room air. $V_{O2}$ and $V_{CO2}$ were measured in WT ($n = 5$) and *mdx/utrn$^{-/-}$* ($n = 5$) mice. For metabolism analysis, groups were classified between 6–7 weeks of age and end-stage disease that was between 8–10 weeks of age depending on disease severity. Fractional concentrations of oxygen ($O_2$) and carbon dioxide ($CO_2$) were measured from the air entering and exiting the plethysmograph chambers (Gas analyzer; AD Instruments, Colorado Springs, CO, USA) similar to that previously described [27] during normoxia and respiratory challenge. Calculation of $V_{O2}$ and $V_{CO2}$ was performed as previously described [27,59]. $V_{O2}$ and $V_{CO2}$ were normalized for body mass (g). $V_{O2}$ and $Vc_{O2}$ were calculated using the average gas concentrations over a 30-second cycle within 5 minutes of baseline and the final 30-second cycle of the respiratory challenge.

## Forelimb grip strength, weight and kyphosis analysis

Forelimb grip strength test, weight and kyphosis were assessed weekly (n = 8 per genotype). Forelimb grip strength was performed as previously described [60] using an Alemno digital grip strength meter (Ahlborn, Holzkirchen, Germany) equipped with a mesh screen. Each mouse was placed on the metal grid and when gripped they were gently pulled by the tail until the grip was released. The meter recorded the maximum force in newtons. The test was repeated three times, and the average was calculated. Kyphosis was assessed and observed while the mouse was freely on a flat surface. The severity of phenotype was graded on a scale from 0–3, being 0 for no pathology and 3 for the worst pathology. The assessment was performed twice and then the average was obtained.

## *Ex vivo* diaphragm muscle function analysis

Diaphragm muscle function was assessed *ex vivo* with a standardized protocol as previously described [61]. At 6 weeks of age, WT ($n = 5$) and *mdx/utrn$^{-/-}$* ($n = 5$) mice were anesthetized with inhaled isoflurane (5% induction, 2–3% maintenance) and subsequently euthanized via double thoracotomy followed by heart removal. The entire diaphragm was removed and placed in Krebs solution (in mM: 137 mm NaCl, 5 mm KCl, 1 mm MgSO4, 1 mm NaH2PO4, 24 mm NaHCO3 and 2 mm CaCl2) aerated with 95% $O_2$/5% $CO_2$. A diaphragm muscle bundle was isolated with a rib and central tendon attached and

suspended vertically between two platinum plate electrodes-the rib sutured to a glass rod and central tendon attached to a Dual-Mode Muscle Lever System (300C-LR; Aurora Scientific, Aurora, ON, Canada), in a water-jacketed tissue bath, containing Krebs solution and continuously gassed with 95% $O_2$/5% $CO_2$ at room temperature. The bundle length was adjusted to achieve maximal tetanic tension (optimal length, $L_O$) and the temperature of the muscle bath was increased to 37°C. After 5 minutes of thermo-equilibrium, the isometric force-frequency relationship was assessed by stimulating the muscle sequentially at 1, 15, 30, 50, 80, 120, 150, and 300 Hz (300ms duration), with a 1-minute interval between stimulation. To determine muscle power, an isotonic contraction was performed at 35% of maximal tetanic force, as previously described [62,63]. Force and shortening length were normalized per cross-sectional area (CSA, N $cm^{-2}$) and optimum length ($L_O$/s), respectively. CSA was estimated by measuring muscle weight and length at $L_O$. Data were analyzed with Dynamic Muscle Analysis Software (version 6, Aurora Scientific, ON, Canada)

### Tissue Processing

Mice were anesthetized with 2–3% inhaled isoflurane and subsequently euthanized via double thoracotomy followed by heart removal. The tongue, diaphragm, and tibialis anterior were harvested and collected. Tissues were embedded and frozen in an optimal cutting temperature (OCT) compound dipped into liquid nitrogen chilled 2-methylbutane cooled to -140 °C for histological analysis. The diaphragm was split into three sections, one was stained for NMJs, another was flash frozen in liquid nitrogen for RNA quantification and the remaining tissue was embedded in OCT for histological analysis. All samples were stored at -80ºC.

### Haematoxylin and eosin staining

OCT-embedded muscles were sectioned at a thickness of 6µm using a cryostat (Leica CM3050 S). The sections were fixed with 4% paraformaldehyde and stained with haematoxylin and eosin as described previously [51]. Images were acquired with the Echo Revolve Microscope using the bright-field light with 40X magnification.

### Picro-Sirus Red staining

OCT-embedded muscles were sectioned at a thickness of 6µm using a cryostat (Leica CM3050 S). Sections were stained for 1 hour with Direct Red 80 (Sigma, 365548 -25G) that was dissolved in 1.3% picric acid (Sigma, P6744), then washed with acetic acid, and dehydrated with 100% ethanol. Images were acquired with the Echo Revolve Microscope using the bright-field light with 40X magnification.

### Dystrophin Immunohistochemistry

OCT-embedded muscles were sectioned at a thickness of 6µm using a cryostat (Leica CM3050 S). The sections were washed with phosphate-buffered saline (PBS) and then blocked in 10% normal horse serum (VWR, 101098-384) for 1 hour. The tissues were incubated with primary antibody (1:100, rabbit polyclonal anti-dystrophin, Thermo Fisher Scientific, PA5-32388) overnight at 4ºC. Then, the tissues were incubated with secondary antibody (1:500, anti-rabbit Alexa Fluor 488, Invitrogen, Carlsbad, CA, USA, A11008) for 1.5 hours and fixed in Formalin (VWR, 10790-714) for 10 mins. Washes with PBS were done between each step. Coverslips were mounted using Vectashield antifade mounting medium with DAPI (VWR, 101098-044). Slides were stored in dark conditions at 4ºC. Images were acquired with the Echo Revolve Microscope using the fluorescence light with 40X magnification.

## Neuromuscular junction immunohistochemistry

Diaphragms were fixed in 2% paraformaldehyde for 15 minutes, then washed with PBS and permeabilized with 2% triton x-100 in PBS for 30 minutes at room temperature, as previously described [55,64]. Tissues were then blocked with 0.5% triton x-100, 2% bovine serum albumin, and 4% normal horse serum in PBS overnight at 4°C. Tissues were incubated with fluorescent primary antibodies anti-mouse Alexa Fluor 488 (1:200, Jackson Immunoresearch laboratories, West Grove, PA, USA, 715-545-150) and anti-chicken Alexa Fluor 647 (1:1200, Invitrogen, Carlsbad, CA, USA, A21449) with Alexa Fluor 594-conjugated α-bungarotoxin (1:1000, Invitrogen, Carlsbad, CA, USA B13423) overnight at 4°C. Finally, tissues were washed with PBS and mounted with Fluoro-gel with Antifade and Fluoro-Gel with Tris Buffer (Electron Microscopy Sciences #17985-71). Images were obtained with a Zeiss 780 upright confocal microscope and maximum intensity projection images were acquired with confocal z-stacks.

**Real-time-qPCR.** 10mg of diaphragm tissues were homogenized in Trizol (Life Technologies, 15596026) with a FastPrep bead homogenizer to extract total RNA [51,52]. Total RNA was then reverse-transcribed (high-capacity cDNA Archive kit, Applied Biosystems, 4368814). To evaluate gene expression, real-time qPCR was performed using TaqMan Gene Expression Master Mix (Applied Biosystems, 439016) and TaqMan primers for *MuSK* (Mm01346929_m1, Life Tech, 4331182)*, and *Chrna7 (*Mm01312230_m1, Life Tech, 4331182)*. Gapdh* was used as a housekeeping gene to normalize the gene expression of the target genes. The data is presented as fold change over control.

## Statistics

Statistical analyses were performed using GraphPad Prism (version 10, La Jolla, CA, USA). Data are presented as mean ± SEM. WBP, metabolism, and behavioral data were analyzed using mixed-model two-way ANOVA followed by a post hoc analysis with uncorrected Fisher's LSD test for multiple comparisons between the groups. Force-frequency relationship was analyzed using two-way ANOVA followed by a post hoc analysis with uncorrected Fisher's LSD test for multiple comparisons between the groups. Expression of mRNA and power in the muscle function of the diaphragm were analyzed using the unpaired Student's *t*-test. Statistical significance was considered at p-value < 0.05 and the following were used to depict statistical significance: * p < 0.05, **p < 0.01, ***p < 0.001, ****p < 0.0001.

## Supporting information

**S1 Data. Raw data for whole body plethysmography, power data analysis and histological quantification.**
(PDF)

## Author contributions

**Conceptualization:** Marán Y. Hernández Rodríguez, Mai K. ElMallah.

**Data curation:** Marán Y. Hernández Rodríguez, Debolina D. Biswas, Aoife D. Slyne.

**Formal analysis:** Marán Y. Hernández Rodríguez, Debolina D. Biswas, Aoife D. Slyne, Jane Lee, Evelyn Scarrow, Sarra M. Abdelbarr, Leonardo F. Ferreira, Mai K. ElMallah.

**Funding acquisition:** Marán Y. Hernández Rodríguez, Charles A Gersbach, Mai K. ElMallah.

**Investigation:** Marán Y. Hernández Rodríguez, Aoife D. Slyne, Jane Lee, Evelyn Scarrow, Sarra M. Abdelbarr, Heather Daniels, Mai K. ElMallah.

**Methodology:** Marán Y. Hernández Rodríguez, Debolina D. Biswas, Aoife D. Slyne, Jane Lee, Heather Daniels, Mai K. ElMallah.

**Project administration:** Marán Y. Hernández Rodríguez, Aoife D. Slyne.

**Supervision:** Ken D. O'Halloran, Leonardo F. Ferreira, Charles A Gersbach, Mai K. ElMallah.

**Validation:** Debolina D. Biswas, Aoife D. Slyne, Evelyn Scarrow, Sarra M. Abdelbarr, Heather Daniels, Leonardo F. Ferreira, Charles A Gersbach, Mai K. ElMallah.

**Visualization:** Debolina D. Biswas, Evelyn Scarrow, Mai K. ElMallah.

**Writing – original draft:** Marán Y. Hernández Rodríguez, Aoife D. Slyne, Jane Lee, Mai K. ElMallah.

**Writing – review & editing:** Marán Y. Hernández Rodríguez, Debolina D. Biswas, Evelyn Scarrow, Ken D. O'Halloran, Leonardo F. Ferreira, Charles A Gersbach, Mai K. ElMallah.

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
