## [Decision Letter · Decision Letter 0]

7 Nov 2024

PONE-D-24-37080Respiratory pathology in the mdx/utrn-/- mice: a murine model for Duchenne Muscular Dystrophy (DMD)PLOS ONE

Dear Dr. ElMallah,

Thank you for submitting your manuscript to PLOS ONE. After careful consideration, we feel that it has merit but does not fully meet PLOS ONE’s publication criteria as it currently stands. Therefore, we invite you to submit a revised version of the manuscript that addresses the points raised during the review process. Your Manuscript has been evaluated by 2 experts in the field. One of them suggested a few minor revisions that I feel will improve the manuscript.

We look forward to receiving your revised manuscript.

Kind regards,

Gerhard Wiche, Ph.D.

Academic Editor

PLOS ONE

“The project was supported by NIH NHLBI PROSPER 5T32HL160494-03 Grant (MHR), NIH NHLBI R01HL171282 (MKE), NIH R01AR069085 (CAG), USAMRMC MD140071 (CAG).”

“The project was supported by NIH NHLBI PROSPER 5T32HL160494-03 Grant (MHR), NIH NHLBI R01HL171282 (MKE), NIH R01AR069085 (CAG), USAMRMC MD140071 (CAG).”

“The project was supported by NIH NHLBI PROSPER 5T32HL160494-03 Grant (MHR), NIH NHLBI R01HL171282 (MKE), NIH R01AR069085 (CAG), USAMRMC MD140071 (CAG).”

4. We note that your Data Availability Statement is currently as follows: [All relevant data are within the manuscript and its Supporting Information files]

Reviewers' comments:

Reviewer's Responses to Questions

**Comments to the Author**

1. Is the manuscript technically sound, and do the data support the conclusions?

Reviewer #1: Yes

2. Has the statistical analysis been performed appropriately and rigorously? 

Reviewer #1: Yes

3. Have the authors made all data underlying the findings in their manuscript fully available?

Reviewer #1: Yes

4. Is the manuscript presented in an intelligible fashion and written in standard English?

Reviewer #1: Yes

5. Review Comments to the Author

Reviewer #1: In their work, M. Hernandez-Rodriguez et al. performed a systematic analysis of the respiratory abilities of mdx/utrn-/- mice, an animal model closely recapitulating the pathology of human patients suffering from Duchenne Muscular Dystrophy (DMD). Respiratory muscle weakness, ultimately leading to respiratory failure, is a serious and life-threatening aspect of this fatal X-linked neuromuscular disorder. Herein, the authors first evaluated life span and development of the pathology (weigth, kyphosis, grip strength), followed by measurements of respiratory function under normoxia (baseline) and after respiratory challenge with hypoxia and hypercapnia. Notably, respiratory muscle function and ventilation was significantly reduced in already in 6-weeks-old mdx/utrn-/- mice and was even more pronounced at end-stage. Finally, the work is complemented by ex vivo evaluation of the functional capacity of the diaphragm and histological analyses (H&E, Pico-Sirius red, immunostaining for dystrophin) of the major respiratory muscle, the diaphragm, as well as of tongue and tibialis anterior, and an evaluation of the neuromuscular junction. The experimental work is well-described and profoundly performed, the manuscript is well formulated. At some points, the authors should provide a more information on how and/or why they performed a certain experiment. Accordingly, there are several minor items that should be addressed:

1. Page 5, line 108: please define the exact age of mice at “end-stage” in the text.

2. Page 6, line 124: please provide a short explanation on why you choose these three muscles.

3. Page 6, lines 130-135: the numbering of Figures 6 and 7 is not correct; the Pico-Sirius red staining is discussed first (Figure 7 according to the figure legends), then the dystrophin immunostaining (Figure 6). Please re-arrange the figures according to the main text.

4. Page 6, lines 136-142: are the neuromuscular junctions in mdx/utrn-/- tongue and tibialis anterior fragmented as well? Why did the author just investigate the diaphragm here?

6. PLOS authors have the option to publish the peer review history of their article (what does this mean? ). If published, this will include your full peer review and any attached files.

**Do you want your identity to be public for this peer review?** For information about this choice, including consent withdrawal, please see our Privacy Policy .

Reviewer #1: No

---

## [Author Response · Author response to Decision Letter 1]

5 Dec 2024

We thank the reviewers for the positive comments and support of the manuscript. We also thank them for their suggestions and believe that these corrections have strengthened the quality of the manuscript. Please see the point-by-point responses to their comments below.

All page and line references are based on the “Revised manuscript with Track changes”.

Reviewer #1:

Minor:

1. Page 5, line 108: please define the exact age of mice at “end-stage” in the text.

We have added this to page 5, line 107, and have also defined this in the the methods section, page 12, line 269-270.

2. Page 6, line 124: please provide a short explanation on why you choose these three muscles.

We provided a short explanation about why we studied the three muscles, page 6 line 125-127.

3. Page 6, lines 130-135: the numbering of Figures 6 and 7 is not correct; the Pico-Sirius red staining is discussed first (Figure 7 according to the figure legends), then the dystrophin immunostaining (Figure 6). Please re-arrange the figures according to the main text.

We have rearranged the figures according to the main text.

4. Page 6, lines 136-142: are the neuromuscular junctions in mdx/utrn-/- tongue and tibialis anterior fragmented as well? Why did the author just investigate the diaphragm here?

For the purpose of this study, we wanted to focus on the respiratory pathology and since the diaphragm is the main inspiratory muscle, we performed ex vivo muscle contraction of the diaphragm and wanted to assess the neuromuscular junction of the diaphragm.

---

## [Editor Report · Decision Letter 1]

10 Dec 2024

Respiratory pathology in the mdx/utrn-/- mouse: a murine model for Duchenne Muscular Dystrophy (DMD)

PONE-D-24-37080R1

Dear Dr. ElMallah,

We’re pleased to inform you that your manuscript has been judged scientifically suitable for publication and will be formally accepted for publication once it meets all outstanding technical requirements.

Kind regards,

Gerhard Wiche, Ph.D.

Academic Editor

PLOS ONE

Additional Editor Comments:

In addressing point 4 of reviewer 1 the authors may want to change the chapter title to "Analysis of diaphragmatic neuromuscular junctions" and streamline the text by combining sentence 1 and 2 (lines141-143).
---

## [Editor Report · Acceptance letter]

PONE-D-24-37080R1

PLOS ONE

Dear Dr. ElMallah,

I'm pleased to inform you that your manuscript has been deemed suitable for publication in PLOS ONE. Congratulations! Your manuscript is now being handed over to our production team.

Kind regards,

on behalf of

Prof. Gerhard Wiche

Academic Editor

PLOS ONE